

# Scale effect challenges in urban hydrology highlighted with a distributed hydrological model

Abdellah Ichiba[1,2], Auguste Gires[1], Ioulia Tchiguirinskaia[1], Daniel Schertzer[1], Philippe Bompard[2], and Marie-Claire Ten Veldhuis[3]

[1]HMCO, Ecole des Ponts ParisTech, Université Paris-Est, 6-8 Av Blaise Pascal Cité Descartes, Marne-la-Vallée, 77455 Cx2, France
[2]Conseil Départemental du Val-de-Marne, Direction des Services de l'Environnement et de l'Assainissement (DSEA), Bonneuil-sur-Marne, 94381, France
[3]Department of Water Management, Faculty of Civil Engineering and Geosciences, Delft University of Technology, PO Box 5048, 2600 GA Delft, the Netherlands

*Correspondence to:* Abdellah Ichiba (abdellah.ichiba@enpc.fr)

**Abstract.** Nowadays, hydrological models are extensively used in urban water management, future development scenario evaluation and research activities. A growing interest is devoted to the development of fully distributed and grid based models, following the increase of computation capabilities. The availability of high resolution GIS information is needed for such models implementation to understand flooding issues at very small scales. However, some complex issues about scaling effects

still remain a serious issue in urban hydrology. The choice of an appropriate spatial resolution is a crucial problem, and the obtained model performance depends highly on the chosen implementation scale.

In this paper we propose a two step investigation framework using scaling effects in urban hydrology. In the first step fractal tools are used to highlight the scale dependency observed within distributed data used to describe the catchment heterogeneity, both the structure of the sewer network and the distribution of impervious areas are analyzed. Then an intensive multi-scale

modeling work is carried out to understand scaling effects on hydrological model performance. Investigations were conducted using a fully distributed and physically based model, Multi-Hydro, developed at Ecole des Ponts ParisTech (Multi-Hydro (2015)). The model was implemented at 17 spatial resolution ranging from 100 m to 5 m.

Results coming out from this work demonstrate scale effect challenges in urban hydrology modeling. In fact, fractal concept highlights the scale dependency observed within distributed data used to implement hydrological models. Patterns of

geophysical data change when we change the observation pixel size. The multi-scale modeling investigation performed with Multi-Hydro model at 17 spatial resolutions confirms scaling effect on hydrological model performance. Results were analyzed at three ranges of scales identified in the fractal analysis and confirmed in the modeling work. In the meantime, this work also discussed some issues remaining in urban hydrology modeling such as the availability of high quality data at higher resolutions and, model numerical instabilities as well as the computation time requirements. But still the principal findings of this paper

allow replacing traditional methods of 'model calibration' by innovative methods of 'model resolution alteration' based on the spatial data variability and scaling of flows in urban hydrology.





## 1 Introduction

Urban environments are very complex systems due to their extreme variability, the interference between human activities and natural processes, and notably the effects of the ongoing urbanization process that changes the land cover and strongly influences the hydrological behavior of urban catchments. Urban hydrological models were developed over the years and used to
simulate the portion of the water cycle in urban environments (Refsgaard and Knudsen (1996); Tech University of Darmstadt and Ostrowski (2002); Salvadore et al. (2015); Hromadka (1987); Daniel et al. (2011); Elliott and Trowsdale (2007); Sarma et al. (1973); Blöschl and Sivapalan (1995)). They can be classified (Salvadore et al. (2015)) based on the nature of the employed algorithms: empirical, conceptual or physically based approaches. They can also be classified based on their spatial resolution and how they represent the complexity of urban hydrology processes (lumped models, semi-distributed and fully-
distributed).

Lumped and semi-distributed models are conceptual ones and often consider a simplified representation of urban catchment heterogeneity: by considering the whole catchment as a unique unit for the first one or by dividing the catchment onto several parts called sub-catchments for the second. These two approaches were largely developed and used for modeling applications because they do not need huge amount of data to be set, they show generally a fast computation time, and they often rely on
a calibration step that forces the model to represent the needed performance. However these models give output information at the sub-catchment scale, which is too coarse for meeting urban water managers requirements in their need to understand some very local flooding problems or to evaluate management strategies at very small scales. Hence the possibility to change spatial resolution of hydrological models gets relevant and several works reported in the literature have investigated this opportunity for sub-catchment models ( Park et al. (2008); Stephenson (1989)). However, the aggregation and desegregation of
sub-catchments changes the model output reflecting a "scale effect" issue and the complex calibration step must be performed again to obtain similar performance as the previous configuration.

The influence of catchment scale on hydrologic response is much more important for fully distributed and gridded-based models, because of their modeling approach that usually consists in representing the high heterogeneity of urban catchment in a gridded format. The choice of an appropriate spatial resolution is always a crucial problem and the obtained model performance
depends highly on the chosen implementation scale. The appropriate spatial resolution is obviously linked to the quality of data available, its spatial resolution and the modeling goal (Dehotin and Braud (2008)). The more accurate representation of the land cover heterogeneity is obtained with high-resolution grid (small pixel size). However, given current computing capabilities and data availability, high-resolution modeling is feasible only for small areas. Therefore, it is important for modelers to understand the effects of spatial resolution in urban hydrologic simulations.
Scale effect and scaling in urban hydrology have been investigated by Gires et al. (2013); Park et al. (2008); Stephenson (1989); Elliott et al. (2009); Wood et al. (1988); Zhang and Montgomery (1994) and reviewed by Blöschl and Sivapalan (1995). Ostrowski (2002) discussed temporal and spatial scaling issues in the context of urban storm-water modeling. Dehotin and Braud (2008) proposed a spatial discretisation methodology applied for distributed hydrological models to get an efficient representation of land cover heterogeneity.




Ghosh and Hellweger (2012) investigated the effects of spatial resolution on predictions of peak flow and total outflow volume in an urban catchment. Zhang and Montgomery (1994) analyzed the DEM grid size and land cover representation, they found that grid size effects influence physically based models, a 10-m grid size provides a substantial improvement over 30- and 90-m data, but 2- or 4-m data provide only marginal additional improvement. Wood et al. (1988) investigated the existence of

a Representative Elementary Area (REA) in the context of hydrologic modeling at the catchment scale.

In this work, we propose to investigate the scale effect in urban hydrology modeling. The investigation will be performed in two steps: in the first part, fractal tools will be used to characterize the scale dependency observed within distributed data (available in commonly used GIS formats) used as input for urban storm models. Then multi-scale modeling investigations

will be carried out using Multi-Hydro model to analyze the effect of this scale dependency on model performance.

## 2   Multi-Hydro model

Multi-Hydro (Fig. 1, Multi-Hydro (2015); Giangola-Murzyn (2013); Ichiba (2016); El Tabach et al. (2009)) is a fully distributed and physically based model developed at Ecole des Ponts ParisTech and has been used by several authors ((Ichiba (2016); Giangola-Murzyn (2013); Versini et al. (2016); Gires et al. (2014)). It is an interacting core between four open source software

packages, each of them representing a portion of the water cycle in urban environments. Multi-Hydro involves a modeling approach that consists in rasterizing the urban domain at a specific spatial resolution chosen by the user. A unique land use class for which hydrological and physical properties are specified is then affected to each pixel.

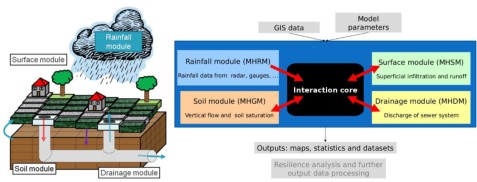

**Figure 1.** Multi-Hydro model is an interacting core between four modules, each of them representing a portion of the water cycle in urban environments. ©Giangola-Murzyn (2013)

The modeling approach involved in Multi-Hydro model relies on solving physical equations that describe the catchment behavior. Seven processes are generally simulated; (1) precipitation, (2) interception and storage, (3) infiltration, (4) overland

flow, (5) sewer flow, (6) infiltration into the subsurface zone and (7) the sewer overflow.

The four modules that make up the core of Multi-Hydro are presented on Fig. 1 :

– The surface component (MHSC) is based on the existing TREX model (Two-dimensional Runoff, Erosion, and Export model) developed by Colorado State University and used in Multi-Hydro only for rainfall-runoff modeling (England





et al. (2007); Velleux et al. (2008)). The surface module computes interception, storage and infiltration occurred at each pixel according to its land use class corresponding properties. The overland flow can occur after satisfying the depression storage threshold, it is governed by the conservation of mass (continuity) and of momentum equations. This flow depends on the surface properties as well as the elevation, and computed using the diffusive wave approximation of

de Saint-Venant equations.

– The rainfall module has been developed at Ecole des Ponts ParisTech, it is used to manage different types of rainfall data (rain gauges, radar data,...) and to process them in the correct input format needed for Multi-Hydro model. The module performs also some data analysis and can be used for radar data downscaling. The data analysis part relies on the Multifractal framework (Lovejoy and Schertzer (1990); Schertzer and Lovejoy (1987)) and has been widely validated by

Gires et al. (2012).

– The Drainage module (MHDC) is based on the 1D SWMM (James et al. (2010)) model (Storm Water Management Model) developed by the US Environmental Agency. It is widely used for urban drainage and modeling purposes. The flow in the sewer network is given by a numerical solution of Saint-Venant equations. This module requires a detailed description of the sewer network (nodes, pipes characteristics, gullies, outlet...).

– The infiltration module relies on VS2DT model developed by the U.S. Geological Survey and it is used to simulate the infiltration into the unsaturated subsurface zone (Healy (1990); Lappala et al. (1987)). This module uses the infiltration depth calculated by the surface module as input, and simulates a 2D infiltration (vertical and 1D horizontal) into the subsurface. This module was not used here because the analysis of the subsurface infiltration was not part of the objectives of this work.

Multi-Hydro is highly demanding on data quality and resolution. Distributed data (usually available in GIS format) describing the topography and land use over the catchment must be collected at a high resolution. Information about the sewer network should also be available, details about all the system components are necessary for the drainage module. Such information are usually available for urban areas and can be obtained from the local authority in charge of the water management. Details about all pipes (geometry, length, diameter as well as inlet and outlet nodes) should be carefully validated as well as all the

system nodes (coordinates and elevation). The subsurface structure must be described as well if there is a need to simulate the infiltration to the unsaturated subsurface zone. The rasterization of the urban domain is the first step of the process of Multi-Hydro implementation. during this process a unique land use class is attributed to each individual pixel. This attribution can be done following at least two methodologies illustrated on Fig. 2. The first one is based on a priority order defined by the user to attribute land use class. The second methodology is based on a majority rule, which means that each pixel will be affected

by the major land class observed within it. These two methodologies were tested and compared by Ichiba (2016). The main findings are reported in Sec.5 (see Fig. 11 and Fig. 12), results indicate that considering the majority rule methodology lead to a better representation of the catchment heterogeneity. Consequently the majority rule was used here during the rasterization step to attribute land class to pixels.





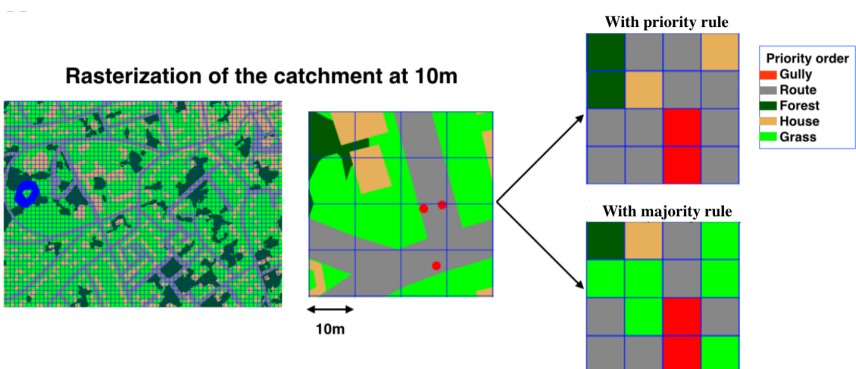

**Figure 2.** Two attribution methodologies are implemented in Multi-Hydro and can be used during the rasterization phase. The first one is based on a priority order defined by the user to attribute land use class whereas the second methodology is based on a majority rule

Multi-Hydro core ensures the connection, interaction, including a retro-action, and data exchange between these four modules after each time loop of 5 min. Indeed, the surface module outputs are used as inputs for the soil and the drainage modules and in the same way the sewer overflow is taken into account on the overland depth for the next step. Multi-Hydro produces a large set of outputs that describe the catchment response. Indeed, the overland water depth maps are available at each time step
as well as the overland discharge flow and the velocity profile at any point of the catchment. Saturation profile of the subsurface zone and the sewer flow are also computed. The model also provides a detailed volume balance at each time step.

Multi-Hydro has already been implemented in several locations for different purposes: cities of Villecresnes (France) and Manchester (UK) for flood mitigation by using barriers and retention basins (Giangola-Murzyn (2013)), Sucy (France) for
retention basin management (Ichiba (2016)), Sevran (France) to study the impact of small scale rainfall variability in urban areas (Gires et al. (2014)), Villepinte and Champs-sur-Marne (France) for large scale implementation of blue and green infrastructures in order to manage storm water (Versini et al. (2016)).

## 3   Case study and data sets

### 3.1   Sucy-en-Brie catchment

The case study presented in this paper is a 2.45 km$^2$ urban catchment located at the Southeast of Paris, in Val-de-Marne County which is a part of the region of Ile-de- France (Fig. 3). The city is connected to Paris with a train at the Sucy-Bonneuil station (30 min travel time to the center of Paris). Known historically as an agriculture area, the city is now highly urbanized with an imperiousness coefficient around 35%. The city is bounded at the north by the Marne river (one of the two main rivers in Paris





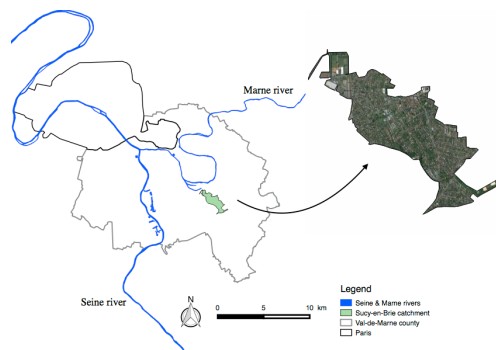

**Figure 3.** Localisation of the Sucy-en-Brie case study in Val-de-Marne County at the Southeast of Paris

region). The area has suffered in the past from several flooding events as a consequence of: (1) the very steep slope (34m/km) that increases water speed and causes overflows in the downstream pluvial network and (2), the increase of imperiousness areas (34%) combined with a soil structure that limits infiltration to the subsurface. The sewer system in this area is a separate one and storm water is routed to the Marne River.

**3.2 Distributed data**

Spatially distributed data is used in this study mainly to set up Multi-Hydro model. It was made available by different institutes in the framework of research collaborations with Ecole des Ponts ParisTech.

– Topography : The Digital elevation model (DEM) of the catchment was obtained from the IGN institute (French National Institute of Forest and Geographic Information), the spatial resolution of the data is 25 m, which is far from meeting the
needs of studies conducted in the framework of this work. Linear interpolation was conducted to obtain data at a better resolution (between 5 m and 10 m).

– Land cover : Figure 4 shows distributed data (available in GIS format) describing the land cover. The data was obtained from the DSEA 94 of Val-de-Marne County (Direction des Services de l'Environnement et de l'Assainissement), its quality is very good with a precision up to 50 cm, but we had to deal with one land use class named "Other" in the
original data, this class was just unknown and introduces missing data. A comparison work with satellite images has been done and the majority of missing data was filled with urban grass.

– Sucy-en-Brie subsurface structure: the subsurface structure was elaborated using data obtained from the BRGM database (Office of Geological and Mining Research) related to the soil investigations done in the past before construction works and archived in the BRGM database. The data indicates that a layer of clay mixed in some places with sand dominates
the majority of the catchment subsurface. In the downstream, near the river, we found a layer of sandy soil, so more





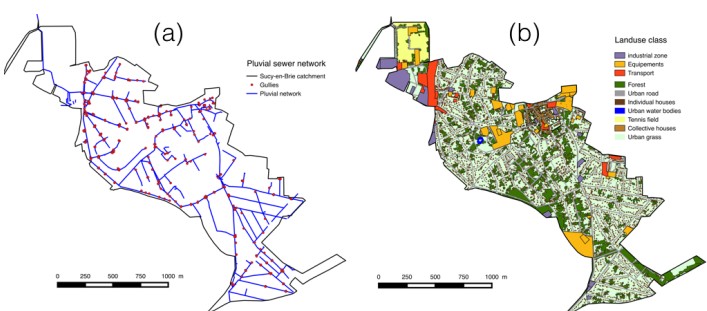

**Figure 4.** The whole pluvial sewer system (a) of Sucy-en-Brie catchment and land use data (b) used to implement Multi-Hydro model

permeable. Physical parameters of soil needed for modeling were obtained from the literature and no measurements were done to verify or to estimate these parameters.

  – Sucy-en-Brie pluvial sewer system : The sewer system in this catchment ( Fig. 4) is a separate one. The pluvial one is routed in the downstream to the Marne river. The DSEA 94 of Val-de-Marne County is the service in charge of the control and the management of the whole system. Data describing the sewer system in this area is a very detailed one. It consists on 2030 nodes and 1015 elements of pipes representing a total length of 25 km, the average slope noticed is around 0.052.

### 3.3 Rainfall and flow measurements data

Rainfall data is provided by the General Council of Val-de-Marne County. The data is coming from a 0.2 mm resolution tipping bucket rain gauge located at the center of the catchment. The data was processed and validated by the DSEA 94 and provided within 5 min resolution. 8 rainfall events (Fig. 5) that occurred between 2010 and 2014 were selected. Their main characteristics are summarized in Table 1. The corresponding flow measurements data was available as well at 5 min resolution. It is coming from a flow sensor located at the outlet of the catchment. The choice was made in this work to use uniform rainfall information in order to focus on the sensitivity if Multi-model to land use variability and to avoid the effect of rainfall spatial variability.

### 4 Methodology

#### 4.1 Part 1 : analysis of the scaling of urban catchment

The first stage of this work is to investigate and identify the scale dependency observed within distributed GIS information (presented in subsection 3.2) used as input for hydrological models. The analysis fully relies on the fractal dimension concept.




| Event | Date | Time start - end | Imax (mm/h) | Total depth (mm) |
|-------|------|------------------|-------------|------------------|
| E1 | 12/06/2010 | 22:00 - 07:00 (+1) | 19.2 | 16 |
| E2 | 12/07/2010 | 06:00 - 14:00 | 24 | 14.2 |
| E3 | 16/07/2011 | 19:00 - 05:00 (+1) | 9.6 | 38.6 |
| E4 | 05/08/2011 | 07:00 - 19:00 | 9.6 | 21.2 |
| E5 | 21/05/2012 | 11:00 - 04:00 (+1) | 43.2 | 19.2 |
| E6 | 08/07/2012 | 01:00 - 09:00 | 21.6 | 11.6 |
| E7 | 08/10/2014 | 06:00 - 15:00 | 21.6 | 33.2 |
| E8 | 12/12/2014 | 18:00 - 18:00 (+1) | 14.4 | 38.6 |

**Table 1.** Main characteristics of the 8 rainfall events selected to perform the scale dependency investigations, Imax is the maximum rainfall intensity recorded in mm/h

Fractal tools are widely used in several science domains including geology, medicine, meteorology and finance (Niu et al. (2016); West (2012); Goldberger and West (1987); Nonnenmacher et al. (2013); Turcotte and Huang (1995); Yanshi and Kaixuan (2002); Turcotte (1989)). In hydrology, the fractal dimension concept has been used in many studies in the past for the various purposes (Mesev et al. (1995); Wu et al. (2013); Thibault and Crews (1995); Frankhauser (1998); Wu and He (2009); Sagar (2004); Jiang et al. (2012); Gires et al. (2013); Radziejewski and Kundzewicz (1997); Gires et al. (2016)).

Fractal geometry was formally introduced by Mandelbrot (1983) and is used to describe geometrical sets that exhibit a great level of complexity, i.e. they are too irregular to be easily described with the help of basic Euclidean concepts but they can be described with the help of simple and repetitive processes. Fractal sets exhibit scale invariance, which means that similar structure will be observed at any scale, the concept of fractal dimension is used to characterize them. In practice, it is computed as follows : the number of pixels $N_{\lambda,A}$ needed to cover the set $(A)$ at a given resolution $\lambda$, which is defined as the ratio between the outer scale $l_0$ and the observation scale $l$ ($\lambda = \frac{l_0}{l}$), follows a power-law:

$$N_\lambda \approx \lambda^{D_f} \tag{1}$$

where the exponent $D_f$ is the fractal dimension and is the asymptotic slope of $N_{\lambda,A}$ vs. $\lambda$ in in log-log plot. The fractal dimension can be therefore defined as follows:

$$D_f = \lim_{\lambda \to +\infty} \frac{ln(N_{\lambda,A})}{ln(\lambda)} \tag{2}$$

Figure 6 shows an example of how fractal analysis is implemented in urban hydrology to analyze a portion of the sewer system. Several pixel size are used to cover the sewer network starting from 2 m pixel size and multiplying the size by two at each step.

The number of pixels $N_\lambda$ needed at a given resolution $\lambda$ to represent the sewer structure is computed and plotted in log-log plot




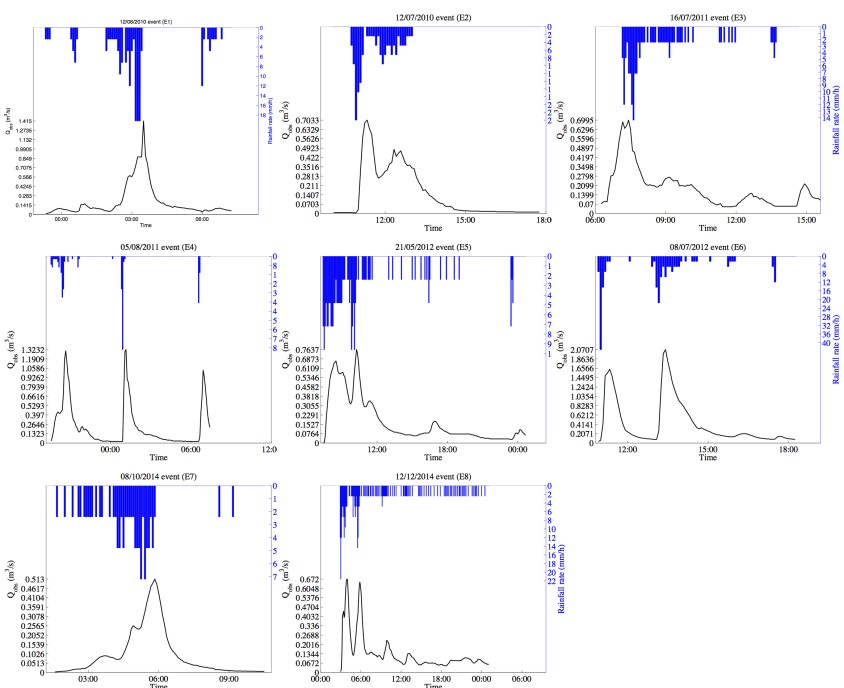

**Figure 5.** Rainfall data and the corresponding flow measurement available for the 8 rainfall events selected to perform the multi-scale modeling investigation.

as function of the resolution $\lambda$ (blue points). From Fig. 6 one can notice the linear behavior retrieved on two separate ranges of scales, which means that the concept of fractal dimension can be use to characterize the sewer network.

In this work, the structure of the urban storm system and the distribution of impervious land use will be analyzed using fractal tools.

### 4.2 Part 2 : Scale effects on fully distributed models outputs

To address the effects of spatial resolution on Multi-Hydro performance, the model was implemented at 17 spatial scales ranging from 100 m to 5 m and intensive modeling works were carried out. Figure 7 shows how the chosen grid size influences the manner land cover heterogeneity is represented in the model. These scale effects will be analyzed with respect to real flow measurements from various points of view according to the performance indicators chosen:




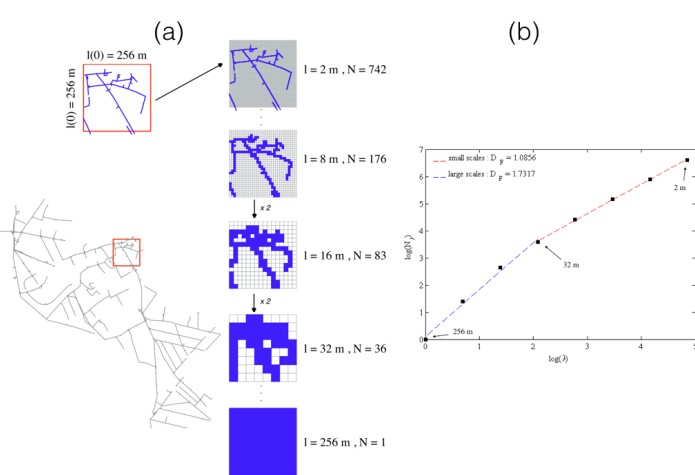

**Figure 6.** Example of Fractal analysis of a portion of the sewer network (256 m size) (a). $N_\lambda$ is plotted as function of $\lambda$ in a log-log plot (b). the scaling behavior was retrieved over two ranges of scales with a break around 32 m

– *Correlation coefficient r :* The correlation coefficient $r$ (Eq. (3)) measures the strength and the direction of the linear relationship between the model output and the observed data, $r$ values range from -1 to +1, positive value of $r$ indicates that the two time series describe the same dynamic (they increase and decrease at the same moment). A correlation greater than 0.8 is generally described as strong, whereas a correlation less than 0.5 is generally described as weak.

$$r = Cor(Q_{mod}, Q_{obs}) = \frac{cov(Q_{mod}, Q_{obs})}{\sigma_{mod} \cdot \sigma_{obs}} \tag{3}$$

– *Nash-Sutcliffe parameter NSE :* Nash-Sutcliffe efficiency coefficient (Eq. (4)) is the most used parameter in urban hydrology to quantify performance of urban models. *NSE* measures how well the model describes the accuracy of model outputs in comparison with a model that only uses the mean of the observed data. *NSE* values range from $-\infty$ to +1. A value of 1 indicates a perfect model, while a value of zero indicates performance no better than simply using the mean. A negative value indicates even worse performance than using just the mean. The efficiency of Nash criterion was discussed in the literature, and many attempts were conducted in order to improve it (Gupta et al. (2009)).

$$NSE = 1 - \frac{\sum_{t=0}^{n}(Q_{obs}^{t} - Q_{mod}^{t})^2}{\sum_{t=0}^{n}(Q_{obs}^{t} - \bar{Q}_{obs}^{t})^2} \tag{4}$$

– *the coefficient of regression $\beta$ :* $\beta$ (Eq. (5)) is computed between each modeled flow $Q_s$ obtained at a given spatial resolution $s$ ($s$ ranges from 100 m to 5 m) and the corresponding flow $Q_{obs}$, it is used here to distinguish spatial scales for which the model overestimates and those for which the model underestimates the observed flow. $\beta$ values range



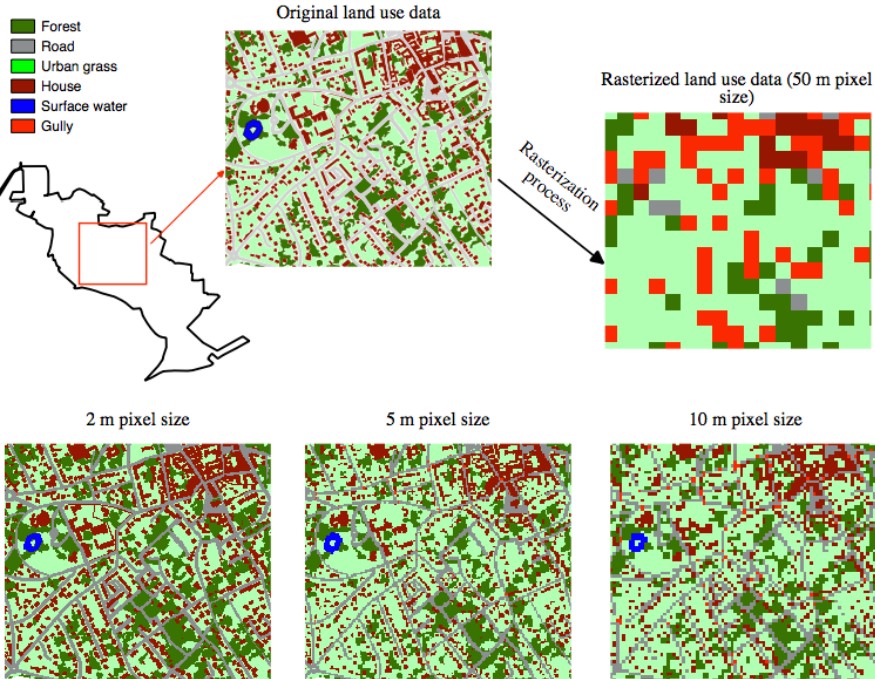

**Figure 7.** Scale effect observed on the catchment land cover. The grid size affects highly the way land cover heterogeneity is considered in the model

between $-\infty$ and $+\infty$, a value of $\beta = 1$ indicates a perfect match between the observed and simulated flows, if $\beta \leq 1$, then the model is underestimating the observed flow, otherwise it is overestimating the observed flow.

$$\beta = \frac{cov(Q_s, Q_{obs})}{var(Q_{obs})} \tag{5}$$

– *Peak flow analysis $\delta r$* : a special focus is given to peak flows. The relative error observed at the peak flow $\delta r$ (Eq. (6)) is used to address effects of scale changes on the modeled peak flow.

$$\delta r = \frac{Q_{mod}^{max} - Q_{obs}^{max}}{Q_{obs}^{max}} \tag{6}$$

In total, 136 simulations (17 spatial resolutions * 8 rainfall events) were run and results were analysed with the help of these parameters. The analysis will help on identifying spatial resolutions for which the model shows good performance with respect to available flow measurements.





## 5 Results and discussions

### 5.1 Fractal analysis of distributed data

#### 5.1.1 Fractal dimension of urban sewer network

Two areas have been selected to perform the fractal analysis for the pluvial sewer system. The purpose of these selection is to

5 minimize the effect of no data pixels by considering two well covered square areas for which the size is a power of 2. Figure 8 shows the 2 m pixel size original data available and the two selected zones. The small area is a 512 m size ($l_0 = 512 = 2^9$ m) and the bigger one is a 1024 m ($= 2^{10}$).

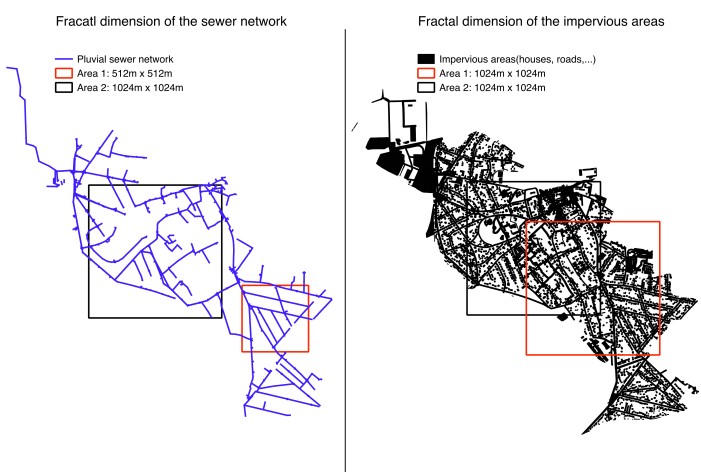

**Figure 8.** The original 2 m pixel size data used to perform the fractal analysis of the sewer system and impervious areas, two well covered areas were selected.

Figure 9 displays results obtained when plotting in a log-log plot the number of pixels $N_\lambda$ needed at a given resolution $\lambda$ to cover the pluvial water system as a function of $\lambda$. Results show a clear adherence to the relation defined in Eq. (1) over

10 two distinct ranges of scales separated by a break at $\approx 64$ m. For small scales (2 m-64 m), the fractal dimension $D_f$ is almost equal to 1 simply reflecting the linear behavior of the sewer pipes structure observed at small scales. For large scales $l \geq 64$ m the fractal dimension $D_f$ is higher than 1.8 (1.82 for area 1 and 1.88 for area 2) suggesting that the pluvial network structure occupies almost the whole 2D space. Thus it becomes difficult at large scales to identify the whole sewer structure (pipes, nodes,...). These results confirm similar conclusions of a multi-catchment work performed in the framework of RainGain




project about fractal analysis of environmental data of 10 pilot sites located in Europe (Gires et al. (2016)). The break at 64 m was related according to this study to the distance between two roads in urban areas.

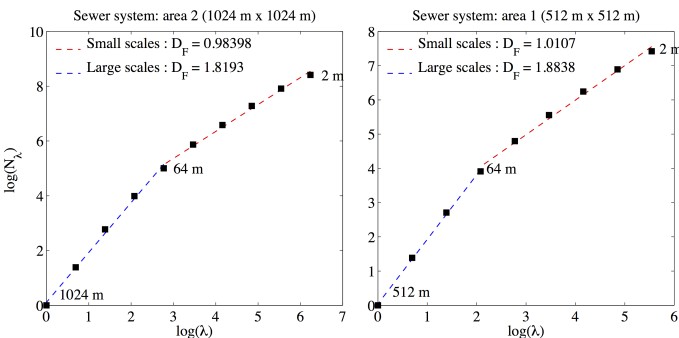

**Figure 9.** Fractal analysis of the sewer structure. Two ranges of scale are identified on both areas; $D_f$ is equal to 1 at small scales (2 m-64 m) and 1.8 for large scales $l \geq 64$ m

### 5.1.2 Fractal dimension of impervious data

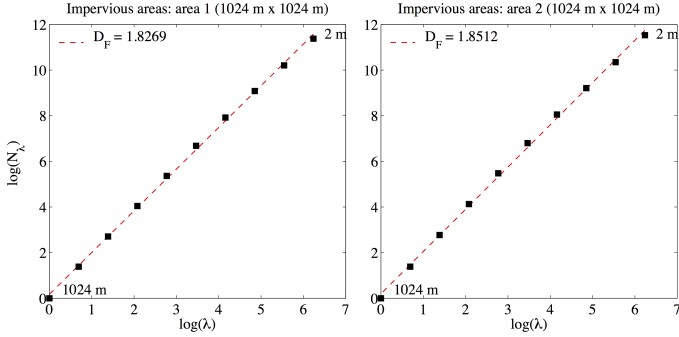

**Figure 10.** Fractal analysis of the impervious data. one unique scaling regime is identified at the whole range of scales available (2 m-1024 m); $D_f$ is greater than 1.8 for both areas.

For impervious data (Fig. 8), two 1024 m size square areas were selected to perform the fractal analysis. Figure 10 shows
5   results obtained. Both areas exhibit a clear and unique scaling regime across the whole range of available scales (2 m-1024 m). The fractal dimension $D_f$ computed is greater than 1.82 (1.82 for Area 1 and 1.85 for Area 2). Results obtained here





show clearly the high scale dependence of urban catchment patterns and demonstrate how important is to well represent urban catchment heterogeneity using gridded models. This scale dependency can have significant consequences on a hydrological model performance.

### 5.1.3 Effect on the urban catchment behavior :

Previous results show that the urban catchment configuration considered in grid-based models highly depends on the scale at which the model is implemented. In fact, spatial patterns observed in the land cover are strongly evolve with the observation scale. Figures 12 and 11 display the distribution of the four main land cover classes (forest, road, grass and house) considered in Multi-Hydro model, as well as the variation of the impervious coefficient $C_{imp}$ (defined as the ratio between impervious surface -gully, roads, houses- and the total surface) as function of the model spatial scale when considering two land cover

pixel attribution methodologies. Results with priority rule are in Figure 11 and those obtained with the majority rule in Figure 12.

Both figures demonstrate that the scale dependency highlighted here is mainly due to the rasterisation methodology followed by Multi-Hydro model during the implementation phase, which affects a unique land cover to each pixel. At small scales both methodologies will lead to the same catchment configuration, whereas results obtained at intermediate scales are different.

The variation of the impervious coefficient $C_{imp}$ (red line) provides an insight into the model behaviour across scale. It is continuously decreasing in the first configuration (Fig. 11), whereas in the second (Fig. 12), the behavior is different. The high values of the $C_{imp}$ coefficient at large scales are due to the fact that the most important soil classes (gully, road and houses) are all impermeable.

When applying the priority rule (Fig. 11), the impervious coefficient $C_{imp}$ is still very high even at high resolution (5m pixels),

meaning that the user must perform hydrological simulations at much more finer resolutions, which is challenging in urban hydrology modelling considering the quality of available GIS data, as well as time consuming needs. In the other hand, Figure 12 shows that the majority rule methodology is more suitable to take into account urban catchment heterogeneity. The land cover distribution is more coherent than the previous methodology. In this case, three ranges of scales can be identified; large scales (100 m-30 m) at which the impervious coefficient decreases significantly from 55% observed in 100 m to its minimum

value of 27% at 30 m, this is due to a great redistribution of land cover classes. At medium scales (30 m-10 m), the impervious coefficient increases from 27% to 37% estimated at 10 m. The second important remark is that for small scales (10 m-5 m), we observe what can be considered as the final configuration of the catchment, the most accurate and closer to the reality on the ground, the impervious coefficient remains stable around 38% which may suggest that the model response will be stable as well at this range of scales.


### 5.2 Scale effects on Multi-Hydro model outputs

For each of the 8 selected rainfall event, 17 simulations were carried out and the corresponding simulated flow time series were retrieved at the outlet pipe where effects are typically smoothed compared to more upstream pipes.




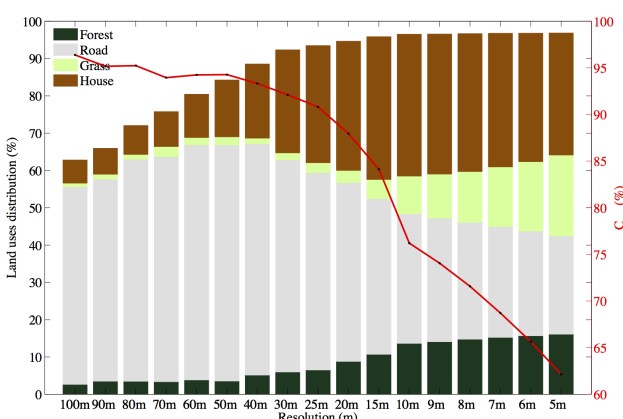

**Figure 11.** Scale dependency observed in the imperviousness coefficient $C_{imp}$ with priority rule as explained in Figure 2. The priority order was set up as follow: gully, road, forest, house, grass.

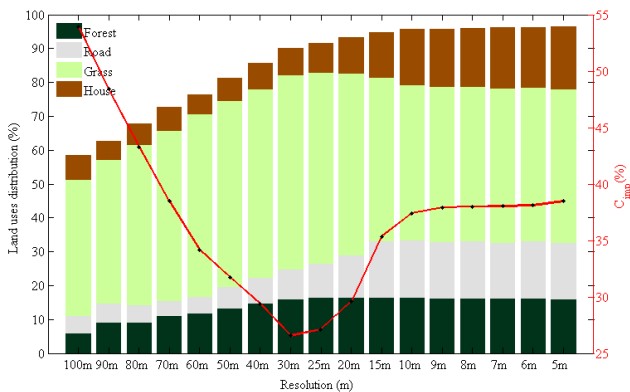

**Figure 12.** Scale dependency observed in the imperviousness coefficient $C_{imp}$ without priority rule as explained in Figure 2

Figure 13 represents all simulated flows $Q_s$ obtained with Multi-Hydro model at 17 spatial scales involved. These results show the high sensitivity of the outputs to the spatial scale of the model.




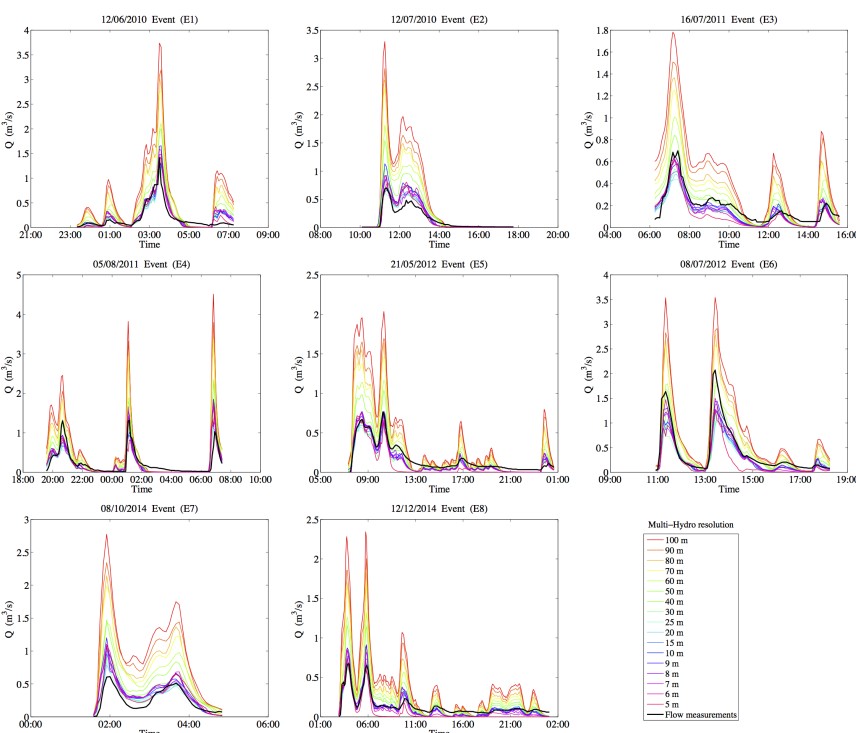

**Figure 13.** Multi-scale modeling outputs compared with observed flow, one can notice the high sensitivity of Multi-Hydro response to the spatial resolution of the model

### 5.2.1 Hydrodynamic evaluation :

The hydrodynamic evaluation aims to quantify how good the model reproduces the flow dynamic observed in the measurements. It is based here on the estimation of the correlation indicator $r$ between modeled and observed data. Boxes presented in the following plots were processed using the 20% and 80% quantiles estimated based on the 8 values (8 black points related to the 8 rainfall events) for each spatial resolution. From Fig. 14, one can notice the high capacity of Multi-Hydro model to reproduce the observed flow dynamic at any spatial scale. In fact, $r$ values range between 0.85 and 0.98 with an average between 0.94 and 0.98 indicating high correlation between modeled and observed data. This trend was also noticed from the graphical comparison (Fig. 13) between modeling outputs and observed data. This demonstrates one of the advantages of physical-based models to reproduce correctly the observed flow dynamic. It also indicates that physical parameters selected from their some-





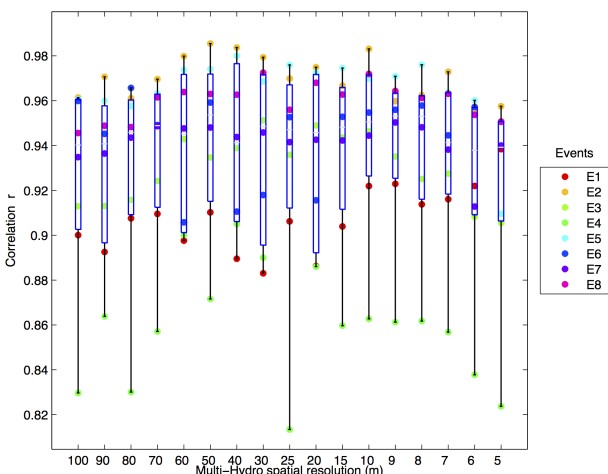

**Figure 14.** Results of model hydrodynamic evaluations; the correlation coefficient r was retrieved for each modeling outputs with respect to real measurements

what representative range and used for the implementation of the model become "correct" as soon as the increase in data resolution creates sufficient spatial variability among these parameters. A variability, which becomes representative (up to the selected precision) of a real urban catchment. This scaling phenomenon produces an alternative to the classical model calibration. The overestimation of the volume is in fact mainly due to an overestimation of impervious areas observed at large scales.

### 5.2.2 Performance evaluation :

The multi-scale performance evaluation of Multi-Hydro model outputs is performed using the three statistical indicator presented in section 4.2; Nash-Sutcliffe efficiency $NSE$, the coefficient of regression $\beta$ and the relative error at the peak flow $\delta r$. Obtained results are summarized in Fig. 15. One can notice the high scale dependency of the obtained model performance

10 which was not the case for dynamic evaluation. In fact all parameters adopt the same trends indicating lower performance at large scales and higher performance at small scales. Model performances are indeed improved as the model resolution increases. From these results, the three ranges of scale previously identified with the help of the fractal analysis (Fig. 11) are also found in Fig. 15. Consequently, performance evaluation will be analyzed at these three ranges of scale.




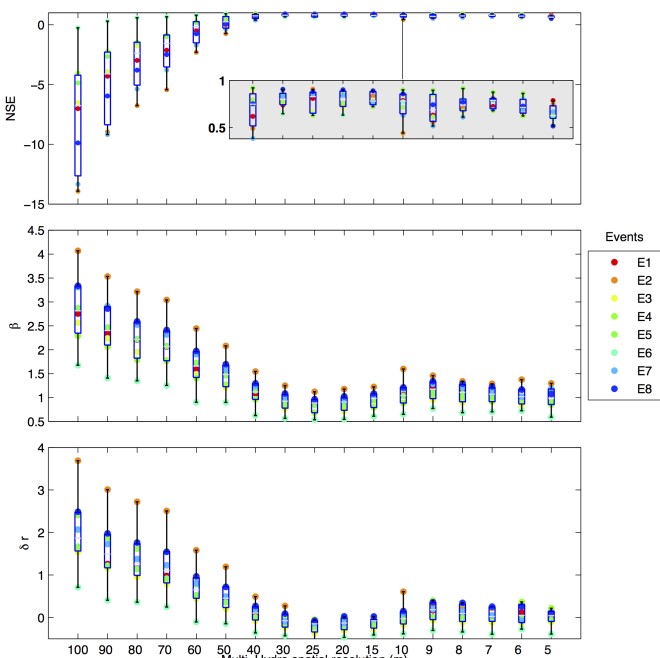

**Figure 15.** Performance indicators *NSE*, $\beta$ and $\delta r$ estimated from Multi-Hydro modeling outputs obtained at the 17 spatial scales with respect to observed data.

| Range of scale | Performance indicators (min/max/mean) | | | |
|---|---|---|---|---|
| | *Correlation* | *NSE* | $\beta$ | $\delta r$ |
| (100 m-40 m) | 0.83/0.99 | -13.92/0.92 | 0.62/4.07 | -0.36/3.69 |
| | /0.93 | /-2.36 | /1.99 | /1.09 |
| (30 m-15 m) | 0.81/0.98 | 0.63/0.91 | 0.54/1.25 | -0.31/0.51 |
| | /0.94 | /0.79 | /0.89 | /0.17 |
| (10 m-5 m) | 0.82/0.98 | 0.44/0.91 | 0.59/1.60 | -0.39/0.61 |
| | /0.93 | /0.72 | /1.06 | /0.19 |

**Table 2.** Min/Max/Mean of performance indicators (*Correlation*, *NSE*, $\beta$ and $\delta r$) calculated at three ranges of scale ((100 m-40 m), (30 m-15 m) and (10 m-5 m))





In Table 2, basic statistics (minimum, maximum and the average) of performance indicators *Correlation*, *NSE*, $\beta$ and $\delta r$ calculated at three ranges of scale (100 m-40 m), (30 m-15 m) and (10 m-5 m) are displayed.

- At large scales (100 m-40 m) : the impervious coefficient $C_{imp}$ of the catchment is very high, it ranges from 45% at 100 m to 30% at 40 m. The model flow obtained at this range of scales exhibits similar dynamic as observed flow. However

performance indicators show very weak performances of the model at this range of scales; *NSE* values range from -13.92 observed at 100 m scale to 0.92 observed at 40 m scale, $\beta$ indicator suggests that the model is highly overestimating observed flow, its values range from 4.07 observed at 100 m scale to a minimum value of 0.62 noticed at 40 m, the average at this range of scales is around 2. In terms of the peak flow analysis, the relative error indicator ($\delta r$) shows clear overestimation of the peak flow at this range of scales up to 369%.

All statistic indicators suggest very weak performances of the model at large scales (100 m-40 m). In fact, the catchment behavior at this range of scales exhibits a high impervious coefficient which means that infiltration is limited and water is routed in majority to the sewer system.

- At medium scales (30 m-15 m) : the model shows its better performances. *NSE* values range between 0.63 and 0.91, with an average around 0.79. The $\beta$ indicator takes values between 0.54 and 1.25, its mean is around 0.89 suggesting a

good fit between modeled and observed data. The relative error indicator ($\delta r$) ranges from -0.31 to 0.51, its mean value is around 0.17 meaning that the model still overestimates the peak flow by 17% on average.

- At small scales (10 m-5 m) : At this range of scales, the model performance remains good but potential trends with regards to scale are unclear. In fact, Table 2 indicates that *NSE* values range between 0.44 and 0.91, its mean value is around 0.72 demonstrating good performance of Multi-Hydro model. The $\beta$ indicator takes values between 0.59 and 1.6,

its mean is around 1.06. The relative error indicator ($\delta r$) ranges from -0.39 to 0.61, its mean value is around 0.19. Slight fluctuations of the model performance are observed at this range of scales and no improvement noticed in the model performance. The model even loses its performance for some rainfall events. These fluctuations observed at this range of scales highlight some specific issues that only take place at this range of scales and influence the model performance. This point will be discussed further in the next section.

### 5.2.3 Specific modeling issues at small scales :

We found it important here to discuss performance of the model in a global framework, especially by taking into consideration some serious problems that one may face when performing high resolution modeling. In fact, as shown in Fig. 15, the model performances are indeed increasing with the spatial scale of the model decreasing, this is due to a better representation of the catchment scaling behavior and the consideration of small scale heterogeneity. However 3 ranges of scales were clearly

identified from previous results. At large scales (100 m-40 m), the model shows a fast computation time (up to only few minutes) but lower performances (the model reproduces the same flow dynamic, but the volume is overestimated by up to 234%). At medium scales (30 m-15 m), the model exhibits high performance (Table 2) and fast computation time. At small scales (10 m-5 m) the urban catchment configuration remains unchanged (the impervious coefficient remains around 37%,



compared to medium scale). However model performances at this range of scales are just unclear and some fluctuations of these performances are noticed. Such fluctuations are in fact related to some non trivial problems that only take place at small scales and should be considered when implementing urban storm models:

1. Quality of distributed data : urban hydrological models in general and fully-distributed ones in particular are highly demanding with respect to distributed data needed for their implementation. A detailed description of the land use is essential as well as distributed topography data. Such data are usually available and can be provided by specialized services. However, its quality is a big issue especially when used to perform high resolution modeling. Two main issues are highlighted here:

   - The spatial resolution of the topography data : the topography is the main driving force for surface water movements and the accuracy of this data has a lot of influence on grid-based models outputs. In our case, the topography data was available at 25 m resolution and interpolation was performed to obtain distributed data at small scales. However, the quality of obtained data below the 25 pixel grid is not fully reliable. The problem is even more striking for small scale down to 2 m (not included in this work, but details could be found in Ichiba (2016)), where the movements of water in the surface were very limited because the elevation gradient becomes very low.

   - Land use description: the land use is also of extreme importance in urban hydrology and specifically for fully-distributed models. In fact, physical properties defined for each pixel depend exclusively on its land use. Such data are usually available especially after huge improvements noticed in the availability of satellite images and new technologies used in this field. However, one commonly and often faced issue of this data is the portion of unknown data, indicating unidentified land use. This is not related to the data resolution, but depends on the processing procedure of satellite and areal images obtained. For the case of Sucy-en-Brie catchment, land use data was available at very good resolution (25 cm), but the portion of unidentified data was about 20% and was filled in most cases by grass.

   At large scales, the problem specific to "no data" pixels has no influence because large pixels usually include a large portion of well-identified land-use classes, like roads and houses. But at small scales, the catchment behavior will be affected by the land use attributed to these no data pixels, and the model response will not be the same if the unidentified areas are filled by grass or by impervious soil.

2. Numerical instabilities : Fluctuation of the model performance noticed at small scales can also be the consequence of numerical instabilities. In fact, the numerical scheme used in Multi-Hydro model for the surface modeling calculations is sensitive to small scale variation and have effect on the model response. Further works should be conducted to better quantify these instabilities.

3. Computation time : it is important in urban hydrology to consider the computation time needed for a model to simulate a given rainfall period. It is in fact one of the first criteria considered by urban water managers for the choice of urban storm models. Fast computation time is even crucial in case of models involved in real-time management processes. For fully-distributed models, the computation time depends on two factors; the size of the catchment and the resolution of the model. For the case of Sucy-en-Brie catchment, Multi-Hydro model shows fast computation time at large scales up





to 10 m (few minutes), and huge computation time is needed at very small scales (5 m-2 m) (several hours). This is due to the numerical scheme, the modeling approach and the huge number and size of the model outputs kept saved for research needs. Improvements should be implemented in the model structure in order to enhance the model performance from this point of view.

4. Mismatch between rainfall input resolution and model resolution: In this study, uniform rainfall input (rain gauge data) was applied to the catchment in all model simulations. Numerous authors have shown that model performance is strongly dependent on rainfall input resolution (Rafieeinasab et al. (2015); Ochoa-Rodriguez et al.; Gires et al. (2015); Ichiba (2016)). Nevertheless, the aim of this study was to investigate the sensitivity of model performance to model resolution independently from rainfall resolution; therefore uniform rainfall was purposefully inputed to the model. Future studies

will look into the combined effects of rainfall and model resolution, based on high resolution rainfall data increasingly available.

5. Interactions between spatial and temporal resolution: In this study, a constant time resolution of 5 minutes was used for rainfall input, flow data and model simulations. Previous studies have shown that a dependency exists between spatial and temporal resolution of rainfall inputs and model simulation results (Rafieeinasab et al. (2015); Ochoa-Rodriguez et al.;

Gires et al. (2015); Ichiba (2016)). Both rainfall phenomena and hydrological processes are exhibit scale dependence, both with respect to their spatial and temporal resolution. Previous studies suggested that a fixed relationship could exist between spatial and temporal resolution and that spatial resolution of rainfall input and model simulation cannot be changed independently of the temporal resolution. Future studies are planned to closely investigate this relationship and the implications this has for hydrological model simulations.

**6   Conclusions**

Results confirm the scale dependency of the obtained model outputs. In fact, the model performances are indeed increasing with the decreasing spatial scale of the model. This is due to a better representation of the catchment small scale heterogeneity, mainly, over the same ranges of scales as for the imperviousness representation. At large scales (100 m-40 m), the model shows a fast computation time (only a few minutes) and also well reproduces the overall flow dynamic, but the flow volumes remain

largely overestimated. At small scales (10 m-5 m) the urban catchment configuration, including the overall imperviousness, becomes scale independent, without any further improvement and even eventual decline of the model performances. The small fluctuations of the model performance at this range of scales are in fact related to some data problems, such as GIS data quality and missing information, as well as to model instabilities, without ignoring the time calculation constraints essential for urban hydrology applications.

Over the remaining medium range of scales (30 m-15 m) for our case study, the model exhibits high performance and fast computation time since the increase in data resolution creates sufficient spatial variability among the grid-based parameters of





the model. Such variability becomes somewhat representative (i.e., up to the selected precision) for the geophysical variability of the studied urban catchment.

Due to a tremendous increase in number of data pixels for grid-based models, one easily understands the difficulty of applying the classical methods for model parameter calibration. Analysis performed here demonstrates that forcing the model

5  to give a better performance by changing its parameters is simply not reasonable for grid-based models because of their strong scale dependency. In turn, such scaling dependency induces an alternative to the classical model calibration. As we demonstrated here, a better consideration of such scale dependency allows defining an optimum range of scales, over which the model performs much better with respect to the measurements. This can be seen as a proposed alternative to the classical parameter calibration of grid-based models.

10  *Acknowledgements.* The Authors acknowledge both the European project INTERREG RainGain (http://www.raingain.eu) and the ANRT association (http://www.anrt.asso.fr) for their financial support of this work. The first author acknowledges the DSEA94 (Direction des Services de l'Environnement et de l'Assainissement) for providing data sets used in this work. A partial financial support of the Chair "Hydrology for resilient cities" endowed by Veolia is greatly acknowledged.





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
