# Peer review of "Scale effect challenges in urban hydrology highlighted with a distributed hydrological model"

_Hydrology and Earth System Sciences, 2017_

## Referee Comment (RC1) · J. Alikhani (Referee) · 16 Jun 2017

General Comment The effect of grid size (referred to as scale) in the hydrology modeling of urban sites is investigated. Two scenarios and 17 grid sizes are considered and the model results are compared to observed flows. The model itself is explained very well with enough supporting materials. The results are statistically analyzed and the performance of the model for each scenario in large, medium, and small scale is obtained based on different factors as well as agreement likelihood and runtime.

The paper is well written with an interesting topic. The paper was successful to discuss its title. It means that a reader can get an actual sense of the scale's effect on the urban hydrology model after reading this paper. Results are presented in informative graphs,

however, in poor quality. Results are also discussed with different parameters that affect the model performance with a clear conclusion.

I just added some minor comments when I was reading the manuscript. These are more in suggestion format. Therefore, it's up to the author to accept, reject, or modify the comments especially those referring to rephrasing.

I vote for a minor revision.

Specific Comments

The abstract is a bit long and it usually should be a single paragraph. I suggest to move the first paragraph to introduction section (if it suits) and merge the second and third paragraphs to a single paragraph.

In Abstract, line 11, don't cite a reference in the abstract section.

Page 2:

Line 15: calibration step should force the model to represent the observed data points not to represent the needed performance. On the general picture, a well-calibrated model can have a low performance depending on the definition.

Line 24: "The choice of an appropriate spatial resolution is always a crucial problem and the obtained model performance depends highly on the chosen implementation scale." -> reference is needed.

Page 3:

Line 2: "they found that grid size effects influence physically based models" -> I would rephrase to: they found that the effect of grid size on the model performance is not linear where . . .

Line 5: give a sentence about their result or remove this line.

Line 12: before dive into the model description, explain a few lies about fully distributed

and physically based models in hydrology. Or do it in the introduction section.

Fig 1: The size and quality of figure 1 are low. It's not readable.

Line 18: what do you mean by "physical equations"

Page 4:

Line 5: Saint-Venant equations -> reference is needed

Page 5:

Fig 2: why in "with majority rule" matrix (in lower right), the pixels in (row:3, col:3) and (row:4, col:4) are red. Shouldn't be gray or yellow, depends on their covered area?

Line 1: "Multi-Hydro core ensures the connection, interaction, including a retro-action, and data exchange between these four modules after each time loop of 5 min." ensures what? Not clear, please rephrase.

First paragraph: The whole paragraph can be reorganized. This is a very messy explanation of a model.

Page 6:

Fig 3: the resolution of the legend is low.

Page 7:

Fig 4: The size and quality are poor.

Line 1: "the literature" -> which literature? Add reference or use another notation

Page 8:

Paragraph 1: move the whole paragraph to the introduction.

Page 9:

Fig 5: The numbers in y and x-axis are hard to read. Generally, the quality of graphs

presented in this manuscript is low. When your paper is published, researchers first look at your graphs to see if that paper worth to download or not. So, I suggest using HD quality graphs always. Merge the second paragraph to the previous one.

Page 10:

Parag 1: No need to explain correlation coefficient and to introduce its equation. It can be found in any statistical textbook.

Page 11:

Line 1: "value of $\beta$ = 1 indicates a perfect match between the observed and simulated" -> not true. The $\beta$ = 1 can show an ideal match, but still, we can have leads square error greater than one.

Page 12:

Line 12: "suggesting that the pluvial network structure occupies almost the whole 2D space" -> why?

Page 15:

Comparing Fig 11 and 12, why the difference between the yellow and gray portion of each bar are significantly different? On the other hand, how come that switch between scenarios can substantially change the portion? It shouldn't be that different.

Page 16:

Line 4: 20% and 80% quantiles -> why not 95% confidence interval (2.5% to 97.5%)

Line 6: R2 isn't a better indicator for model accuracy evaluation?

Page 20:

Line 1: some fluctuations of these performances are noticed -> how can performance fluctuate? It's not clear. Maybe oscillation is a better word here!
Conclusion:

Please first explain about the goal of the study. The conclusion section should be a summary of your study plus the result. You just dived in the results section.

Technical Correction

title:

I think scale's effect is a better word. But I'm not sure if you want to change that entire the manuscript. "Scale's effect on the urban hydrology ..."

Page 2:

Line 2-4: break down the sentence to two.

Line 7: Move the (Salvadore et al. (2015)) to the end of the sentence.

Line 11: Please add at least one reference for each type: (lumped models, semi-distributed and fully distributed)

22: change "much more important" to "more important"

Line 25: "The appropriate spatial resolution is obviously linked to the quality of data available, its spatial resolution and the modeling goal", the "its spatial resolution" is redundant.

27: "is obtained" -> can be obtained.

30: been investigated by ... -> been investigated by researchers, for example ...

Page 3:

Line 2: DEM -> Use full form before first application of any abbreviation

Line 3: effects -> affects

Page 4:

Line 12: Environmental Agency -> Environmental Protection Agency

Line 20: demanding -> demanded

Line 27: unique land use class -> unique class of land use

Line 29: majority rule -> rule of majority

Page 6:

Line 2: imperiousness or impervious?

Line 3: is a separate one and storm water -> is a separate from storm water

Line 14: very good -> high

Page 7:

Line 3: a separate one -> a separate distribution system from storm runoff system

Line 3: use "system" instead of "one"

Page 9:

Line 9: is represented -> as represented

Page 19:

Line 13: the model shows its better performances -> the model shows a better performance OR the model shows its best performance

Line 14: values between 0.54 and 1.25, its mean is around 0.89 -> values between 0.54 and 1.25 with a mean around 0.89

Line 17: the model performance remains good -> the model performance remains high

---

## Referee Comment (RC2) · Anonymous Referee #2 · 11 Jul 2017

General Comments

It is disappointing that this seven author paper has not been better prepared for peer review. A simple comparison with the related excellent paper by Gires et al. (2017) in HESS, applying similar fractal analyses to semi-distributed models for a variety of catchments, illustrates what might have been achieved if the seven authors had combined their respective strengths to prepare the paper thoroughly for peer review. This takes time and requires attention to the detail of presentation. I would encourage the authors to do just that in a resubmission.

Some pointers are:

1. Introduce equations as part of a sentence and define symbols as they are introduced.

2. Comply with the preparation guidelines of HESS: Table caption at top etc.

3. The English should be improved to the standard of Gires et al. (2017). The abstract provides a good example of where improvement is needed. Punctuation, plurality (data are) and use (omission) of the definite article requires attention, for example.

4. Be considerate of the peer reviewers' task. Prepare a paper fit for external review through a process of thorough internal review. It should be a delight to read like Gires et al. (2017).

Now turning to the science contribution of the paper.

The paper complements and goes beyond that by Gires et al. (2017) by focussing on a single urban catchment (Sucy-en-Brie) and using a model (Multi-Hydro) that is distributed.

A key feature investigated through fractal analysis is how different spatial properties (land use, impervious cover, sewer structure) are introduced into a distributed hydrological model configuration at different scales (model resolutions) and how this impacts performance. Priority and Majority rules are compared: obviously this can make a big difference to model response as a function of scale (model resolution) and this is demonstrated for impervious cover. The authors do not consider alternatives to these rules to make the formulation more scale invariant such as a fractional approach. This deserves some comment.

The paper seems more motivated by creating and observing multifractal behaviour than addressing the modelling problem it presents: some more discussion of the latter would be good. Also, more detail needs to be given on how these properties function within the model to gain clearer insight and understanding (for example, the precise meaning of imperviousness coefficient needs to be understood in terms of model function).

The science of the paper, and the practical application addressed, is of interest and
deserving of publication in HESS. However, the paper presentation requires thorough revision followed by re-review.

Some more detailed comments follow.

Detailed Comments

P1 Very brief guidance on improving the abstract follows (as an example). "Hydrological models are...activities. There is a growing interest in the development..such model implementation. ...crucial problem, and model performance...Both the structure...modeling investigation is...17 spatial resolutions. Results demonstrate scale. The fractal concept...with the Multi-Hydro model...and confirmed through modelling. This work also discussed...requirements. The principal findings..."

P2 huge amounts of data...hydrological models becomes relevant..aggregation and disaggregation...is obtained using a high-resolution grid...

P3 ...representation. They found... physically-based models: a 10 m ...used to configure urban storm models.....the multi-Hydro...assigned to each pixel

P5, 6 imperviousness coefficient

P7 ...sensitivity of the Multi-model to land use...(should this be the Multi-Hydro model???)

P10, 11 These equations need to be introduced properly as part of a sentence and terms defined as they are introduced.

P10, 11 and elsewhere. More care needs to be taken with the word "parameter". NSE is Nash-Sutcliffe Efficiency (a performance metric or statistic, not a parameter).

P11 The inequality is wrong.

P12 The purpose of this selection...

P14 imperviousness coefficient ???

[Figure]

P14 Section 5.1.3 heading – remove : (and elsewhere)

P14 which assigns a unique land cover

P14 land use classes (not soil)

P17 . . .spatial variability among these properties. (not parameters)

P17 In fact all indicators reveal a similar trend of higher performance at smaller scales. (Not parameters.)

P19 Improve style of "We found it important here. . ."

P23 Dehotin et al. is no longer in HESS Discuss.

P23 Gires et al. now published in HESS 2017.

P24 Rafieeinasab et al. (2015) – correct reference to Journal of Hydrology

P24 Thibault and Crews (1995) – correct reference to "Flux, 19, 17-30, 1995."

P25 Yanshi and Kaixuan – Change title to mixed case.

---

## Author Comment (AC1) · 31 Aug 2017

First of all, authors would like to thank the two referees for agreeing to spend time reviewing this paper. Their thoughtful remarks and suggestions enabled to enrich the paper and bring other information and details that make it more straightforward. We fully accept their comments and suggestions and all requested modifications were considered and the text was updated accordingly.

Thank you also for reporting in details in your remarks (line indications, suggestions…) which has made the correction work easier.

Please find below, a point-by-point response to your comments and also an indication of changes made on the paper based on your feedback.

**J. Alikhani (Referee)**

jamal.alikhani@gmail.com

General Comment The effect of grid size (referred to as scale) in the hydrology modeling of urban sites is investigated. Two scenarios and 17 grid sizes are considered and the model results are compared to observed flows. The model itself is explained very well with enough supporting materials. The results are statistically analyzed and the performance of the model for each scenario in large, medium, and small scale is obtained based on different factors as well as agreement likelihood and runtime.

The paper is well written with an interesting topic. The paper was successful to discuss its title. It means that a reader can get an actual sense of the scale's effect on the urban hydrology model after reading this paper. Results are presented in informative graphs, however, in poor quality. Results are also discussed with different parameters that affect the model performance with a clear conclusion.

I just added some minor comments when I was reading the manuscript. These are more in suggestion format. Therefore, it's up to the author to accept, reject, or modify the comments especially those referring to rephrasing.

I vote for a minor revision.

Specific Comments

The abstract is a bit long and it usually should be a single paragraph. I suggest to move the first paragraph to introduction section (if it suits) and merge the second and third paragraphs to a single paragraph.

In Abstract, line 11, don't cite a reference in the abstract section.

The Abstract was updated following your feedback.

*Page 2:*

*Line 15: calibration step should force the model to represent the observed data points not to represent the needed performance. On the general picture, a well-calibrated model can have a low performance depending on the definition.*

The referee is indeed correct and the sentence was changed accordingly in the updated manuscript.

*Line 24: "The choice of an appropriate spatial resolution is always a crucial problem and the obtained model performance depends highly on the chosen implementation scale." -> reference is needed.*

This actually was one of the main findings of the first author's PhD work, following several modelling investigations performed using both a fully distributed (Multi-Hydro) and a semi-distributed one (CANOE). Both models show different performance when changing the implementation scale. The reference to this PhD thesis was added
.

*Page 3:*

*Line 2: "they found that grid size effects influence physically based models" -> I would rephrase to: they found that the effect of grid size on the model performance is not linear where . . .*

The change was made in the text to clarify the sentence.

*Line 5: give a sentence about their result or remove this line.*

As suggested by the referee, a short sentence describing their work was added to the text

*Line 12: before dive into the model description, explain a few lies about fully distributed and physically based models in hydrology. Or do it in the introduction section.*

*Fig 1: The size and quality of figure 1 are low. It's not readable.*

The size was change to benefit fully from the initial quality. It's now clearer.

*Line 18: what do you mean by "physical equations" Page 4:*

The expression "physical equations" as it was cited in the text refers to the fact that Multi-Hydro model relies on solving physical equations that describe the hydro-dynamic processes of the catchment, instead of relying on some conceptual models simply trying to relate an input to an output.

*Line 5: Saint-Venant equations -> reference is needed Page 5:*

A reference added to the text. The reader can find there more details about the diffusive wave approximation of de Saint-Venant equations used in TREX model to compute the surface flow.

*Fig 2: why in "with majority rule" matrix (in lower right), the pixels in (row:3, col:3) and (row:4, col:4) are red. Shouldn't be gray or yellow, depends on their covered area?*

Thank you for your question, this allowed us to improve the text as well as in the caption to clarify this point. Actually in both methodologies, the gully class remains priority to ensure the connection between the surface and the drainage modules which explain why the pixels you cited remain red.

*Line 1: "Multi-Hydro core ensures the connection, interaction, including a retro-action, and data exchange between these four modules after each time loop of 5 min." ensures what? Not clear, please rephrase.*

The sentence was rephrased as follow:
The four modules are connected via the Multi-Hydro core, which groups together a set of codes allowing interaction, retro-action (feedback) and data exchange between these modules. In this case study, these interactions are performed after each time loop of 5 min

*First paragraph: The whole paragraph can be reorganized. This is a very messy explanation of a model.*

As suggested by the referee, the whole paragraph was re-organized in the updated version of the manuscript.

*Page 6: Fig 3: the resolution of the legend is low.*

The size of the figure was increased. This makes the legend clearer.

*Page 7: Fig 4: The size and quality are poor.*

Fig 4 was improved. Thank you for your careful reading

*Line 1: "the literature" -> which literature? Add reference or use another notation*

We added a reference to the table used in Multi-Hydro model to estimate physical parameters for 11 types of soil.

*Page 8: Paragraph 1: move the whole paragraph to the introduction.*

Done, thank you. As suggested by the referee, the whole paragraph was moved to the introduction section.

*Page 9: Fig 5: The numbers in y and x-axis are hard to read. Generally, the quality of graphs presented in this manuscript is low. When your paper is published, researchers*

*first look at your graphs to see if that paper worth to download or not. So, I suggest using HD quality graphs always. Merge the second paragraph to the previous one.*

All Figures, and notably Fig. 5, were improved following your feedback
*Page 10:*

*Parag 1: No need to explain correlation coefficient and to introduce its equation. It can be found in any statistical textbook.*

We would prefer to keep the precise explanation of all static metrics used in the investigation to avoid any confusion. If the referee insists, they could easily be removed.

*Page 11:*

*Line 1: "value of β = 1 indicates a perfect match between the observed and simulated" -> not true. The β = 1 can show an ideal match, but still, we can have leads square error greater than one.*

Thank you, your suggestion was considered, and text modified accordingly.
*Page 12:*

*Line 12: "suggesting that the pluvial network structure occupies almost the whole 2D space" -> why?*

What we simply wanted to say is that the fractal dimension found between 1.82 and 1.88 (for the large scales, i.e. $l \geq 64$ m) was close to the dimension of the embedding space equal to 2. This means that on this range of scales, the structure of the pluvial networks fills most of the space. This feature is expected from such a network in dense area since citizen expect falling water to be evacuated. This was clarified in the text.

*Page 15:*

*Comparing Fig 11 and 12, why the difference between the yellow and gray portion of each bar are significantly different? On the other hand, how come that switch between scenarios can substantially change the portion? It shouldn't be that different.*

This is related to the methodology applied for the land use class attribution. Both methodologies lead to different configuration of the urban catchment. Thanks to the referee's comment, a clarification was added to the text (P.16) with an example explaining the difference of the portion of road pixels observed between the two figures for 100 m pixel size.

"Both figures demonstrate that the scale dependency highlighted here is mainly due to the rasterisation methodology performed in Multi-Hydro model during the implementation phase, which assigns a unique land cover to each pixel. At very small scales both methodologies will

basically lead to the same catchment configuration, whereas results obtained at intermediate scales are different. To illustrate these differences, let us consider the case of pixels of size 100m. In an urban environment it is very likely that such a pixel will intersect a road. Then, with priority rule, since "road" pixels have a high level of priority, this will make the portion of pixels affected with road land use class greater. This portion decreases as the pixel size decreases. On the other hand, with the majority rule (Fig. 13), the portion of road pixels is smaller because the roads will usually not occupy the greater portion of such pixel."

*Page 16:*

*Line 4: 20% and 80% quantiles -> why not 95% confidence interval (2.5% to 97.5%)*

The box plots are obtained from the computation of 8 samples corresponding to 8 rainfall events. All the results obtained are plotted and no information was removed. The boxes corresponding to the 20% and 80% quantiles were added only with an indicative purpose. This was clarified in the manuscript.

*Line 6: R2 isn't a better indicator for model accuracy evaluation?*

We tried in this paper to use a set of statistics to address the model performance from different point of view. So we used the correlation coefficient which is quite similar to the R2 coefficient.

*Page 20:*
*Line 1: some fluctuations of these performances are noticed -> how can performance fluctuate? It's not clear. Maybe oscillation is a better word here!*

This refers to the fact that the trend observed in statistics as a function of pixel size for large and medium scales (the improvement of all statistics as the pixel size decreases) in no longer valid at small scales where an fluctuations of statistics are noticed (they increase at 10m and 9m before decreasing at 7m). This was clarified in the manuscript.

*Conclusion:*

*Please first explain about the goal of the study. The conclusion section should be a summary of your study plus the result. You just dived in the results section.*

The conclusion was updated following your remark. A paragraph was added to explain the scope of the study.

*Technical Correction*

*title:*

*I think scale's effect is a better word. But I'm not sure if you want to change that entire the manuscript. "Scale's effect on the urban hydrology . . ."*

We would rather stick to the original title. Actually, we believe that the clarifications following the referees comments enable to make the current title more straightforward with regards to the updated content.

*Page 2:*

*Line 2-4: break down the sentence to two.*

This was modified in the text.

*Line 7: Move the (Salvadore et al. (2015)) to the end of the sentence.*

This was modified in the text.

*Line 11: Please add at least one reference for each type: (lumped models, semi-distributed and fully distributed)*

Reference for each type of modelling approach was added to the text. Thank you

*22: change "much more important" to "more important"*

This was modified in the text.

*Line 25: "The appropriate spatial resolution is obviously linked to the quality of data available, its spatial resolution and the modeling goal", the "its spatial resolution" is redundant.*

This was changed in the text.

*27: "is obtained" -> can be obtained. 30: been investigated by . . . -> been investigated by researchers, for example . . . Page 3: Line 2: DEM -> Use full form before first application of any abbreviation Line 3: effects -> affects Page 4:*

This was modified in the text. Thank you

*Line 12: Environmental Agency -> Environmental Protection Agency Line 20: demanding -> demanded Line 27: unique land use class -> unique class of land use Line 29: majority rule -> rule of majority*

This was modified in the text. Thank you

*Page 6:*

*Line 2: imperiousness or impervious?*

This was modified in the text. Thank you very much

*Line 3: is a separate one and storm water -> is a separate from storm water*

The sentence was modified as follow to clarify the text:
The drainage system in this area is a separate one (i.e. is a separate network for the waste water system and the storm water system). The storm runoff system is routed to the Marne River.

*Line 14: very good -> high*

This was modified in the text.
*Page 7:*

*Line 3: a separate one -> a separate distribution system from storm runoff system*

*Line 3: use "system" instead of "one"*

The sentence was modified in the text as mentioned in the previous page.
*Page 9:*

*Line 9: is represented -> as represented*

I guess that is represented is fine but I might be mistaken.
*Page 19:*

*Line 13: the model shows its better performances -> the model shows a better performance OR the model shows its best performance*

The sentence was modified in the text
*Line 14: values between 0.54 and 1.25, its mean is around 0.89 -> values between 0.54 and 1.25 with a mean around 0.89*

The sentence was modified in the text
*Line 17: the model performance remains good -> the model performance remains high*

The sentence was modified in the text

---

## Author Comment (AC2) · 31 Aug 2017

First of all, authors would like to thank the two referees for agreeing to spend time reviewing this paper. Their thoughtful remarks and suggestions enabled to enrich the paper and bring other information and details that make it more straightforward. We fully accept their comments and suggestions and all requested modifications were considered and the text was updated accordingly.

Thank you also for reporting in details in your remarks (line indications, suggestions…) which has made the correction work easier.

Please find below, a point-by-point response to your comments and also an indication of changes made on the paper based on your feedback.

**Anonymous Referee #2**

*General Comments*

*It is disappointing that this seven author paper has not been better prepared for peer review. A simple comparison with the related excellent paper by Gires et al. (2017) in HESS, applying similar fractal analyses to semi-distributed models for a variety of catchments, illustrates what might have been achieved if the seven authors had combined their respective strengths to prepare the paper thoroughly for peer review. This takes time and requires attention to the detail of presentation. I would encourage the authors to do just that in a resubmission.*

*Some pointers are:*

*1.Introduce equations as part of a sentence and define symbols as they are introduced.*

This was updated in the text following your remark, especially for page 10 and 11.

*2. Comply with the preparation guidelines of HESS: Table caption at top etc.*

The HESS guidelines were checked and updates were made.

*3.The English should be improved to the standard of Gires et al. (2017). The abstract provides a good example of where improvement is needed. Punctuation, plurality (data are) and use (omission) of the definite article requires attention, for example.*

The paper was carefully proofread to improve the English.

*4. Be considerate of the peer reviewers' task. Prepare a paper fit for external review through a process of thorough internal review. It should be a delight to read like Gires et al. (2017).*

As suggested by the referee, we carefully did it for the revised manuscript.

*Now turning to the science contribution of the paper.*

*The paper complements and goes beyond that by Gires et al. (2017) by focussing on a single urban catchment (Sucy-en-Brie) and using a model (Multi-Hydro) that is distributed.*

*A key feature investigated through fractal analysis is how different spatial properties (land use, impervious cover, sewer structure) are introduced into a distributed hydro-logical model configuration at different scales (model resolutions) and how this impacts performance. Priority and Majority rules are compared: obviously this can make a big difference to model response as a function of scale (model resolution) and this is demonstrated for impervious cover. The authors do not consider alternatives to these rules to make the formulation more scale invariant such as a fractional approach. This deserves some comment.*

The referee raises an interesting point. It was actually thought of during the PhD of the first author that served as a basis for this paper. Indeed, the possibility to implement other rules was investigated, but the model formulation allows only one land use class (characterized by few parameters such as the conductivity) per pixel. It means that implementing a fractional approach as suggested by the referee would require to strongly increase the number of classes as well as to develop a multifractal spatial characterization of key parameters such as conductivity. Those are possible motivating future investigation paths but they are outside the scope of the current study. Hence it was chosen to limit the study to two rules for affecting pixels'class; while keeping the number of classes reasonable.

As suggested by the referee, this point was clarified in the manuscript.

*The paper seems more motivated by creating and observing multifractal behaviour than addressing the modelling problem it presents: some more discussion of the latter would be good.*

In fact, the paper was motivated by the fact that on the one hand the inputs of the hydrological models exhibit scale invariant features while on the other hand distributed models are implemented at a single resolution.
Hence the question we tried to investigate in this paper is "at which resolution should we implement the model?"; keeping in mind practical constraints such as missing data at high resolution or longer computation time. The main goal of the paper is to investigate the existence and try to identify the appropriate resolution (or a range of resolution) for Multi-Hydro implementation. To this hand we first used fractal analysis (not multifractal) to analyze the features of the inputs of the model and in a second part we performed multi-scale modelling work.

As these points were raised by the referee, this was added in the introduction section, and highlighted in the conclusion section to make more explicit the goal of the paper.

*Also, more detail needs to be given on how these properties function within the model to gain clearer insight and understanding (for example, the precise meaning of imperviousness coefficient needs to be understood in terms of model function).*

The imperviousness coefficient is actually not a parameter of the modelling formulation. It is simply a quantity used to gain some insight on the inputs of the model and how its overall features changes with resolution. It refers here to the proportion of those impervious pixels (road, building…), i.e. the ones that participate directly to the rapid runoff. A precision was added to the text page 16 to make this clearer.

*The science of the paper, and the practical application addressed, is of interest and deserving of publication in HESS. However, the paper presentation requires thorough revision followed by re-review.*

*Some more detailed comments follow.*

*Detailed Comments*

*P1 Very brief guidance on improving the abstract follows (as an example). "Hydrological models are...activities. There is a growing interest in the development..such model implementation. . . .crucial problem, and model performance. . .Both the structure. . .modeling investigation is. . .17 spatial resolutions. Results demonstrate scale. The fractal concept. . .with the Multi-Hydro model. . .and confirmed through modelling. This work also discussed. . .requirements. The principal findings. . ."*

Thank you for your detailed comments and suggestions. The text was updated.

*P2 huge amounts of data...hydrological models becomes relevant aggregation and disaggregation. . .is obtained using a high-resolution grid. . .*

This was modified in the text.

*P3 . . .representation. They found. . . physically-based models: a 10 m . . .used to configure urban storm models. . ...the multi-Hydro. . .assigned to each pixel*

This was modified in the text.

*P5, 6 imperviousness coefficient*

This was modified in the text. Thank you

*P7 ...sensitivity of the Multi-model to land use...(should this be the Multi-Hydro model???)*

Yes, the necessary modifications have been implemented. Thank you

*P10, 11 These equations need to be introduced properly as part of a sentence and terms defined as they are introduced.*

The text was updated following your remark.

*P10, 11 and elsewhere. More care needs to be taken with the word "parameter". NSE is Nash-Sutcliffe Efficiency (a performance metric or statistic, not a parameter).*

The referee is indeed right and this was changed.

*P11 The inequality is wrong. P12 The purpose of this selection. . . P14 imperviousness coefficient ???*

This was modified in the text. Thank you

*P14 Section 5.1.3 heading – remove : (and elsewhere)*

This was modified in the text.
*P14 which assigns a unique land cover*

This was modified in the text.

*P14 land use classes (not soil)*

This was modified in the text.

*P17 . . .spatial variability among these properties. (not parameters)*

This was modified in the text.

*P17 In fact all indicators reveal a similar trend of higher performance at smaller scales. (Not parameters.)*

This was modified in the text.

*P19 Improve style of "We found it important here. . ." P23 Dehotin et al. is no longer in HESS Discuss. P23 Gires et al. now published in HESS 2017. P24 Rafieeinasab et al. (2015) – correct reference to Journal of Hydrology P24 Thibault and Crews (1995) – correct reference to "Flux, 19, 17-30, 1995." P25 Yanshi and Kaixuan – Change title to mixed case.*

This was corrected in the text; thank you for your careful reading.